# Tumor Angiocrine Signaling: Novel Targeting Opportunity in Cancer

**DOI:** 10.3390/cells12202510

**Published:** 2023-10-23

**Authors:** Victor Oginga Oria, Janine Terra Erler

**Affiliations:** Biotech Research and Innovation Centre (BRIC), Faculty of Health and Medical Sciences, University of Copenhagen, Ole Maaløes Vej 5, DK-2200 Copenhagen, Denmark; janine.erler@bric.ku.dk

**Keywords:** angiocrine factors, angiocrine signaling, endothelium

## Abstract

The vascular endothelium supplies nutrients and oxygen to different body organs and supports the progression of diseases such as cancer through angiogenesis. Pathological angiogenesis remains a challenge as most patients develop resistance to the approved anti-angiogenic therapies. Therefore, a better understanding of endothelium signaling will support the development of more effective treatments. Over the past two decades, the emerging consensus suggests that the role of endothelial cells in tumor development has gone beyond angiogenesis. Instead, endothelial cells are now considered active participants in the tumor microenvironment, secreting angiocrine factors such as cytokines, growth factors, and chemokines, which instruct their proximate microenvironments. The function of angiocrine signaling is being uncovered in different fields, such as tissue homeostasis, early development, organogenesis, organ regeneration post-injury, and tumorigenesis. In this review, we elucidate the intricate role of angiocrine signaling in cancer progression, including distant metastasis, tumor dormancy, pre-metastatic niche formation, immune evasion, and therapy resistance.

## 1. Introduction

Cancer is recognized as an evolutionary and ecological system involving continuous, dynamic, and reciprocal interactions between cancer cells and the surrounding tumor microenvironment (TME) [1,2,3]. The TME comprises cellular components such as endothelial cells, fibroblasts, immune cells, stem cells, and acellular factors such as the extracellular matrix, cytokines, growth factors, and hormones. The TME plays a pivotal role in tumor initiation, progression, and metastasis due to the dynamic and persistent crosstalk between cancer cells and other cell types in the TME [3,4,5]. Therefore, characterizing the TME is essential in cancer research and the cornerstone of current drug development. 

As aberrant organs, tumors require a steady supply of blood vessels to provide nutrients and oxygen and remove metabolic waste products. Endothelial cells line the inner vascular system that separates the circulating blood from tissues. Thus, the vascular endothelium has multifunctional roles, including angiogenesis, thermoregulation, leukocyte migration, proliferation, permeability, homeostasis, and control of vasomotor tone. As critical components of the TME, endothelial cells are involved in angiogenesis, which refers to the formation of new blood vessels from pre-existing vessels [4,6,7]. Angiogenesis is a tightly orchestrated mechanism in physiological processes such as embryonic development and wound healing. In cancer, pathological angiogenesis is driven by the overexpression and secretion of pro-angiogenic factors. This creates an imbalance between anti-angiogenic and pro-angiogenic factors, which favors the recruitment of new vascular networks. Some key pro-angiogenic factors include vascular endothelial growth factor (VEGF), fibroblast growth factor (FGF), interleukin 8 (IL-8), epidermal growth factor (EGF), and angiopoietins (ANGPTs) [6,8]. 

## 2. Tumor Angiogenesis and Resistance to Anti-Angiogenic Therapies

As the tumor grows, the proximity to peripheral blood vessels reduces, leading to low pH, severe hypoxia, and nutrient deprivation. This hypoxic TME upregulates the transcription and secretion of soluble angiogenic and growth factors, which activates different tumor survival pathways [5,8,9]. These factors induce the proliferation and spouting of peripheral endothelial cells. Unlike normal physiology, this process is highly dysregulated and develops disorganized, immature, leaky, and tortuous blood vessels, resulting in poorly perfused tumors resistant to therapy [4,6,8]. The improper development of the tumor vasculature is partly caused by high levels of soluble factors secreted by tumor cells, of which VEGF plays a key role. This poor functionality of the tumor vasculature has a profound effect on the TME, which may lead to sustained hypoxia, decreased infiltration of immune cells, and subsequent metastasis [5,8,9,10]. The normalization of the tumor vasculature emerged as a therapeutic strategy to correct this faulty phenotype. Tumor vascular normalization restores proper vessel perfusion and oxygenation, limiting tumor invasiveness and improving cancer therapy effectiveness. This led to the development and approval of therapeutic monoclonal antibodies against VEGF and kinase inhibitors against VEGF receptors, as VEGF is the most well-characterized angiogenic factor [8,9,11]. 

VEGF is overexpressed in most solid tumors and is a crucial mediator of tumor angiogenesis [8,9,11]. In 2004, the Food and Drug Administration (FDA) approved the use of bevacizumab, a humanized monoclonal antibody against VEGF-A, for untreated metastatic colorectal cancer. Subsequent clinical trials confirmed the effectiveness of bevacizumab in additional solid tumors, including renal cell carcinoma, breast cancer, cervical cancer, non-small cell lung cancer, and ovarian cancer [9,11]. Given the initial success of anti-VEGF therapies, a second strategy was developed to target the VEGF signaling pathway by inhibiting its receptors VEGFR1, VEGFR2, and VEGFR3, as well as other receptors regulating angiogenesis. This effort led to the development and FDA approval of eleven anti-angiogenic therapies, including sorafenib, sunitinib, pazopanib, axitinib, and vandetanib for the treatment of different solid tumors (Table 1) [8,9,11]. 

Despite the initial successes of these anti-angiogenic therapies, the efficacy of these therapies in clinical settings remains limited. This is because of the development of inherent or adaptive resistance towards these therapies. Tumors may mediate this resistance by activating angiogenic signaling independent of VEGF-A, cancer–stromal cell interactions, tumor endothelial cell heterogeneity, and physiological processes like vessel co-option and vascular mimicry [9,12,13,14,15]. Some clinical investigations support these hypotheses. The most straightforward theory of resistance to anti-angiogenic therapies is the compensatory upregulation of alternative pathways. Anti-VEGF therapies have been reported to increase the secretion of alternative factors such as VEGF-D, ANG-2, PDGF, PIGF, VEGF-C, and SDF1-α in the blood of cancer patients [8,12,16]. These factors bind to their receptors on endothelial cells and activate alternative pathways that promote angiogenesis. The co-targeting of these alternative pathways and anti-angiogenic therapies is currently being tested in clinical settings. For example, the dual targeting of PDGF and VEGF has been tested in phase I/II trials with acceptable toxicities and promising efficacy [17]. 

Second, a study involving clear-cell renal-cell carcinoma (ccRCC) patients showed that treatment with VEGFR inhibitors correlated with an increase in intra-tumoral cancer-associated fibroblasts (CAFs), subsequently promoting lymph node dissemination and resistance to these therapies [18]. The recruitment of local and distal stromal cells, such as bone marrow-derived cells and cancer-associated fibroblasts, may also mediate resistance to anti-angiogenic therapies. These stromal cells upregulate and secrete factors such as CXCR4, PIGF, G-CSF, and SDF1-α that activate the alternative angiogenic pathways and promote the infiltration of recruited stromal cells [14,19,20]. Third, tumor resistance to anti-angiogenic therapies may be attributed to vessel co-option. This is a non-angiogenic vascularization mechanism where tumors hijack the pre-existing vascular network of the non-malignant tissue they colonize [5,21,22]. During this process, tumor cells migrate along the outer surface of pre-existing blood vessels and infiltrate the tissue space, ultimately incorporating these vessels into the tumor. This way, tumors co-opt existing blood vessels to secure their nutrients and oxygen supply without stimulating angiogenesis [21,22]. Nevertheless, the molecular mechanisms behind vessel co-option remain poorly understood and are currently a research subject.

Resistance to anti-angiogenic therapies may also be linked to tumor endothelial cell heterogeneity. Single-cell transcriptomic studies of lung cancer [15], breast cancer [23], glioblastoma [24], and colorectal cancer [25] reveal that the tumor endothelium is characterized by a high level of plasticity, which is the driving force behind pathological angiogenesis and remodeling of the vascular niches. While anti-angiogenic therapies were developed because pathological angiogenesis relied on VEGF/VEGFR signaling, bulk and single-cell transcriptomic studies reveal a complex heterogeneity of the tumor vasculature, which should guide the design of rational therapeutic regimens [26]. For example, the single-cell sequencing of TECs in glioblastoma revealed five distinct endothelial phenotypes associated with endothelial cell activation and remodeling of the vascular niche [24]. While peripheral endothelial expressed genes are associated with quiescence and blood–brain barrier integrity, endothelial cells embedded in the GBM core overexpressed genes regulating angiogenesis, basement membrane remodeling, tip cell proliferation, cytoskeleton rearrangement, and partial BBB integrity [24]. In glioblastoma, breast, and colorectal tumors, there was also an enrichment of alternative angiogenic factors, including biglycan, ANGPT2, HPSG2, KDR, COL4A1, COL4A2, VWA1, and TIMP1, which may likely compensate for VEGF blockade [23,24,25]. A similar single-cell study that investigated the heterogeneity of lung TECs identified sixteen previously unknown phenotypes, including basement membrane breaching, immune cell recruitment, and antigen presentation [15]. Interestingly, proliferating tip TECs comprise less than 10% of lung tumor TECs, raising the question of whether targeting such a small population inhibits tumor angiogenesis [15,26]. A deeper understanding of the molecular basis of this complex TEC heterogeneity will likely provide insights into developing novel vascular-specific therapies. 

## 3. Angiocrine Signaling

The resistance to anti-angiogenic therapies has been primarily attributed to factors other than endothelial cells. Yet, a growing body of literature suggests that endothelial cells are not passive in the TME, merely responding to exogenous cues from neighboring cells [2,27]. The role of endothelial cells has grown beyond angiogenesis. Emerging consensus demonstrates that endothelial cells actively participate in the TME, secreting different factors (Figure 1), hereby termed angiocrine factors, instructing their local microenvironment [2,27]. Examples of angiocrine factors include cytokines, chemokines, ECM components, growth factors, and trophogens that act in a juxtacrine or paracrine manner to regulate their microenvironments. This type of endothelial-mediated instructive signaling is known as angiocrine signaling and was first proposed by Shahin Rafii in 2010 [28]. Angiocrine signaling is not novel to the growing tumor. Since its inception, research has demonstrated that angiocrine factors play crucial roles during early developmental processes, organogenesis, organ regeneration, and homeostasis [29,30]. 

Organ-specific endothelial cells actualize their functions based on their adaptability to specific microenvironments. Endothelial cells in capillary beds of specific organs secrete specific angiocrine factors that act in a paracrine manner to regulate organ response to fibrosis, inflammation, immune response, regeneration, and tissue repair [31]. For example, angiocrine signaling is an essential regulator of bone development. Bone formation begins with the migration and localization of cells in specific micro-niches, followed by condensation of mesenchymal cells. Even though the process is devoid of vascular invasion, peripheral endothelial cells secrete angiocrine factors such as transforming growth factor-beta 1 (TGF-β1) and connective tissue growth factor (CTGF) that regulate mesenchymal condensation [31,32]. Disruption of this molecular communication between the vascular system and skeletal structure leads to different pathological conditions such as osteoporosis, nonunion fractures, and avascular necrosis [31,32,33]. The muscle is a complex tissue containing resident stem cells known as satellite cells necessary for postnatal muscle growth, repair, and regeneration [34]. These satellite cells express VEGFA, which recruits peripheral endothelial cells, enhancing juxtacrine signaling between these two cell populations. The cellular proximity stimulates the expression of delta-like protein 4 (DLL4) on endothelial cells, which binds to its receptor on satellite cells, inducing quiescence [31,34]. Genetic deletion of VEGFA in muscle satellite cells or using anti-VEGF inhibitors reversed this phenotype, demonstrating the significance of angiocrine signals in this context. 

## 4. Targeting Angiocrine Signaling for Cancer Therapy

Under physiological conditions, endothelial-derived angiocrine factors coordinate and regulate organ development, homeostasis, and regeneration [28,29]. Evidence shows that tumors can exploit these functions to generate microenvironments that support tumor proliferation, immunosuppression, stemness, dormancy, and invasiveness. Therefore, direct targeting of angiocrine factors or combinatorial treatment modalities presents a promising platform for developing effective therapeutic strategies. 

## 5. Metastasis

Vascular endothelial cells secrete different angiocrine factors implicated in invasion and distant metastasis. For example, crosstalk between endothelial cells and tumor cells accelerates metastatic potency in triple-negative breast cancer (TNBC) [35]. A pro-inflammatory environment induces the tumor secretion of plasminogen activator inhibitor-1 (PAI-1), stimulating the expression and secretion of chemokine CCL5 from endothelial cells, which acts in a paracrine fashion on TNBC cells to facilitate metastasis (Figure 2A). Importantly, blocking of CCL5 using a neutralizing antibody attenuates PAI-1 secretion, suggesting a positive feedback loop between endothelial and TNBC cells [35]. Endothelial cells in human carcinomas frequently display activated Notch 1 receptors. Notch signaling regulates cellular crosstalk during organ development and oncogenesis [36]. In their study, Wieland and others extensively describe the role of tumor endothelial cells Notch1 signaling in lung metastasis. Endothelial Notch1 activation in the primary tumor induces two essential metastatic programs. First, it promotes endothelial cell senescence in the primary tumor, facilitating tumor cells’ intravasation [36]. Second, it also co-stimulates Notch1 activation in endothelial cells in the pre-metastatic niche, which induces the expression of cell adhesion molecule VCAM1 and neutrophil infiltration, facilitating the homing of metastasizing tumor cells in the lungs [36]. Treatment with VCAM1 and Notch 1 receptor blocking antibodies ablated Notch-driven metastasis in a mouse model of ovarian carcinoma [36]. Further studies using different metastasis models are needed to elucidate the role of endothelial Notch1 activity as a therapeutic target. 

Pancreatic ductal adenocarcinoma (PDAC) is an aggressive tumor and highly resistant to many therapeutic agents, including gemcitabine, one of its approved first-line chemotherapies [37]. PDAC is a hypovascularized tumor characterized by low vascular density, partly due to the dense surrounding stroma, contributing to therapy resistance. A recent study using different PDAC mouse models investigated the duality of endothelial cell focal adhesion kinase (FAK) expression and gemcitabine treatment in distant metastasis [37]. In a spontaneous metastasis model, the combination of endothelial-FAK loss and gemcitabine treatment did not affect primary tumor growth or tumor angiogenesis. However, it significantly reduced liver and lung metastasis, prolonged gemcitabine-treated mice’s survival, and down-regulated MAPK, Akt, and RAF signaling [37]. This is consistent with previous studies that showed endothelial FAK expression enhancing vascular integrity and barrier function [38,39], underlining the importance of FAK signaling in metastasis. These results were validated in PDAC patients, where low levels of endothelial-FAK were associated with increased survival and reduced relapsed post-gemcitabine treatment, underlying its clinical utility [37]. Other angiocrine factors implicated in tumor metastasis include ANGPT2, IL-8, CXCL16, and CCL2 in liver cancer [40]; IL-6, TGF-β, and MMP9 in prostate cancer [41]; CXCL5 in head and neck cancer [42]; CXCL1 in gastric cancer [43]; IL-6, G-CSF, GM-CSF, and sICAM1 in colorectal cancer [44]; and lysyl oxidase (LOX) in renal carcinoma [45], all of which have been implicated in metastasis. To assess their therapeutic potential, further pre-clinical and clinical investigations are required using existing and novel pharmacological agents against some of these factors (ANGPT2, IL-6, IL-8, TGF-β, and LOX). 

## 6. Therapy Resistance

Previous studies in hematopoietic malignancies indicate that the endothelium plays a prominent role in tumor growth and response to therapy [46,47,48]. Interactions between endothelial cells and T-cell acute lymphoblastic leukemia (T-ALL) cells in patient-derived xenograft (PDX) models modulate reactions to drugs with anti-leukemic activity. While individual T-ALL xenografts are sensitive to different anti-cancer compounds, co-culture with endothelial cells lowers T-ALL sensitivity, likely attributed to angiocrine factors secreted by endothelial cells [46]. Studies involving acute myeloid leukemia (AML) show that tumor cells activate resting endothelial cells to generate an inflammatory endothelium, which subsequently induces the adhesion of specific leukemia cells by the overexpression of E-selectin [47]. The adhered leukemia cells enter a quiescent state, which shields them from chemotherapy agents, and they later detach to induce AML relapse via an endothelial-derived IL-8 mechanism [47,49]. Other studies have also demonstrated that the endothelial-derived granulocyte-stimulating factor (G-CSF) induces a pro-inflammatory microenvironment, which promotes the clonal evolution of pre-leukemic stem cells to leukemic stem cells [50]. Studies in B cell lymphoma show that tumors stimulate the neighboring endothelium via fibroblast growth factor 4 (FGF4) to upregulate the Notch ligand Jagged 1 (JAG1). Upregulation of JAG1 on endothelial cells activates the Notch2-HEY1 pathway in lymphoma cells, driving extra nodal metastasis and chemoresistance [48]. Inducible deletion of FGF receptor 1 or JAG1 in endothelial cells in lymphoma cells diminishes disease aggressiveness and prolongs mouse survival [48]. Angiocrine factors are also implicated in therapy resistance in solid tumors. Overexpression and secretion of CXCL1 by breast cancer cells in the TME recruit CD11b^+^Gr1^+^ myeloid cells, which produce S100A8/9 that promote survival in metastatic sites [51]. Targeting these tumors with chemotherapy triggers the secretion of TNF-α from endothelial cells, which heightens the secretion of CXCL1 and amplifies the CXCL1-S100A8/9 loop, leading to chemoresistance [51]. Endothelial-derived FGF2 activates the Akt/mTOR signaling pathway to promote cancer proliferation and chemoresistance to docetaxel, a first-line chemotherapy for castration-resistant prostate cancer [52]. 

## 7. Establishment of Pre-Metastatic Niches

The formation of pre-metastatic niches (PMN) is a complex process involving selective remodeling of secondary organs by tumor- and stromal-derived factors. Growing evidence shows that endothelial-derived factors play a significant role in PMN formation, and their therapeutic targeting could be clinically beneficial to patients [53]. Sunitinib, a VEGFR inhibitor, has been used to treat different solid tumors but has been unsuccessful in managing metastatic breast cancer (MBC) [54]. The molecular mechanisms behind MBC’s resistance to sunitinib have been unclear, yet patients undergoing this therapy develop distant metastasis. Other studies demonstrated that MBC’s resistance to sunitinib was due to increased breast cancer stem cells in the TME, the activation of vascular mimicry, and the upregulation of hypoxia-response genes [55]. Here, sunitinib was reported to create a PMN-like microenvironment in MBC by inducing senescence in endothelial cells in both the primary tumor and distant metastatic organs (Figure 2B) [56]. In this study, the authors showed that sunitinib induced endothelium senescence and activated the secretion of inflammatory chemokines and VCAM1, which recruited neutrophils, macrophages, and myeloid cells that synergistically established the PMN. Endothelial senescence was also associated with decreased vascular endothelial cadherin, which led to loose endothelial cell junctions promoting the transmigration of tumor cells through the endothelial barrier to the PMN [56], validating an observation reported in an earlier study [10]. This could likely explain MBC’s resistance to anti-angiogenic therapies. 

Breast cancer metastasis involves both hematogenous and lymphatic dissemination. Organ-residing lymphatic vessels serve as metastatic routes to distant sites, pre-modeling the PMN. Paracrine signaling between breast tumor cells and lymphatic endothelial cells (LECs) induces angiocrine signaling that selectively modifies the PMN (lungs and lymph nodes) to direct breast cancer metastasis [57]. In this study, the authors show that LECs residing in the lymph nodes and lungs are conditioned by tumor-secreted IL-6 to express and secrete CCL5 in a p-STAT3-HIF-1α-VEGF-dependent manner, creating an inflammatory PMN for metastasizing cells. In addition, these tumor-educated LECs induce angiogenesis in their resident organs, facilitating tumor cell extravasation and colonization of the PMNs [57]. Similar findings were reported in mouse models of Lewis lung carcinoma, where endothelial-derived ANGPT2 recruited metastasis-associated macrophages induced a pro-inflammatory and angiogenic response for PMN formation [58]. In both studies, pharmacological agents against IL-6, the CCL5/CCR2 signaling axis, and ANGPT2 quenched the inflammatory phenotype and inhibited metastatic outgrowth [57,58]. The downregulation of endomucin (EMCN), a transmembrane protein expressed on the surface of the endothelium, significantly correlates with metastasis and tumor recurrence in lung cancer patients [59]. Endothelial-specific EMCN knockout (EMCN^ecko^) mice had a higher lung metastasis burden compared to their wild-type counterparts, an observation that was independent of primary growth. Pathologically, EMCN^ecko^ mice had elevated Ly6G^+^ neutrophils in the pre-metastatic lung, which, upon polarization, promoted the remodeling of PMN via TGF-β. Inhibition of TGF-β activity attenuated neutrophil polarization and lung metastasis [59]. 

## 8. Tumor Dormancy

Angiocrine factors have also been implicated in regulating tumor dormancy, a critical stage of disease development where the tumor is quiescent. Breast cancer is the most commonly diagnosed cancer in women globally. While there has been an improvement in treatment and survival over time, the risk of metastatic relapse from dormant tumors is high [60]. The microenvironment at the metastatic site regulates the establishment and maintenance of the dormancy–metastatic outgrowth cycle. Studies involving mice and human clinical samples show that micro clusters of disseminated tumor cells (DTCs) may persist in a long-term state of dormancy before awakening and colonizing the metastatic site [60,61]. Multiple theories, including angiocrine signaling, have been postulated as a mechanism to explain the dormancy–metastatic outgrowth phenomenon. In mouse models of breast cancer metastasis, cues from the microvascular niches regulate the sleep–wake processes of DTCs residing in the bone marrow, lungs, and brain, and this partly depends on the status of the endothelium. Using organotypic models of bone marrow and lung microvascular niches, a recent study demonstrated that the fate of breast cancer cells depended on the status of endothelial cells (Figure 2C) [61]. Quiescent tumor-associated endothelium secreted thrombospondin-1 (TSP1) that established a dormant niche to sustain the inactivity of invasive breast tumor cells [61], a phenotype that was also reported in another study of breast cancer dormancy [62]. Furthermore, TSP-1 co-localized with dormant DTCs on lung vasculature in both experimental and spontaneous mouse metastasis models. On the contrary, sprouting neovasculature in organotypic and in vivo models promoted micrometastatic outgrowth through periostin and TGF-β1 secreted by endothelial tip cells [61]. While these results partly elucidate the mechanisms behind dormancy, they open up the possibility of additional angiocrine factors involved in regulating this process. 

Bone metastasis increases with age, making the bone microenvironment a common metastatic site for many solid tumors and is associated with poor prognosis and significant morbidity [63]. As aging profoundly affects the bone microenvironment, there is evidence that this unique environment activates micrometastatic outgrowth from resident DTCs [64]. Vascular niches in the bone secrete unique age-associated secretomes that promote quiescence or DTC proliferation. A comparison of the bone secretome between young and aged mice showed an increase in the secretion of quiescence-promoting factors, including Bmp6, Bmp7, Dkk1, Dkk3, Thbs2, and Tgfb2, by pericytes and endothelial cells residing in the bone microenvironment of young mice. These factors enable DTCs in young mice to be resistant to chemoradiation, a phenotype lost in aged mice, making them susceptible to metastatic outgrowth [64]. In addition, these quiescence-promoting factors that are secreted in the bone microenvironment of young mice were also upregulated in normal bones and those in metastatic tumor cells after chemoradiation in both aged and young mice [64]. Interestingly, the susceptibility of old mice to chemoradiation was also associated with a loss of bone-specific PDGFRβ^+^ pericytes and a reduction in PDGF-BB secreted by aging endothelial cells in these niches [64]. These results demonstrate the regulatory role of angiocrine factors that regulate the status of DTCs in the bone microenvironment. Similar experimental models of dormancy have also shown that the bone marrow endothelial niche is responsible for the awakening of dormant ER+ breast cancer cells through ANGPT2 signaling [65]. In patient-derived xenograft models of colorectal cancer and T-ALL, the expression of Notch ligand DLL4 in endothelial cells was reported to facilitate an escape from dormancy and metastatic colonization by activating Notch3 signaling in tumor cells. Treatment with an anti-DLL4 neutralizing antibody or the silencing of Notch3 in tumor cells had marked pro-apoptotic, anti-proliferation, and reduced tumor burden in vivo [66].

## 9. Immune Evasion

Angiocrine factors have been known to regulate the immune milieu of the TME and dictate the course of the disease. Infiltrating immune cells are critical components of the TME and, depending on their phenotype, are used as prognostic indicators of disease progression [2,4,27]. The tumor endothelium is crucial in the recruitment of immune cells to inflamed tissues (Figure 3A). Activated leukocytes induce a pro-inflammatory transcription program in endothelial cells, leading to the secretion of nitric oxide that promotes vasodilation and facilitates immune-cell trafficking [67]. The transendothelial migration of immune cells during trafficking is further regulated by CCL2 and CXCL10 secreted by the activated endothelium [67]. Evidence shows that macrophages support glioblastoma growth, invasion, and therapeutic resistance by secreting different factors that regulate the brain TME [68]. Immune evasion in glioblastoma, which partly contributes to therapeutic resistance, relies on the spatiotemporal regulation of macrophage activation. A previous study illustrated that crosstalk between tumor-promoting macrophages and the proximate vasculature creates a permissive niche for angiocrine-induced macrophage polarization driven by interleukin 6 (IL-6) and colony-stimulating factor 1 (CSF-1) (Figure 3B) [69]. Mechanistically, IL-6 and CSF-1 synergistically activate the Akt-mTOR pathway in macrophages, inducing the expression of PPARγ and HIF-2α, which promote arginase-1 expression and subsequent macrophage polarization [69]. These results were validated in vivo, where endothelial-specific deletion of IL-6 inhibited macrophage polarization and improved the survival of GBM-bearing mice, illustrating a vascular-dependent pathway for GBM immune evasion [69]. Targeting IL-6 for cancer therapy remains an unexploited niche despite the central role of IL-6 in tumor inflammatory responses. Clinical evidence suggests that therapeutic blocking of IL-6 has been successful in managing rheumatoid arthritis and Castleman disease [70]. However, most of the developed anti-IL-6 monoclonal antibodies have failed clinical trials against solid tumors, including pancreatic cancer, renal cell carcinoma, melanoma, prostate cancer, and colorectal cancer, and thus, they have never been approved for clinical use [71]. 

Crosstalk between immune cells and endothelial cells determines the fate of anti-tumor immunity and the efficacy of immunotherapy [2,6,28,72,73]. As an angiocrine factor, VEGF promotes an immunosuppressive microenvironment through several mechanisms. A previous study demonstrated that the VEGF secreted by endothelial and colorectal cancer cells enhanced the expression of negative immune checkpoints PD-1 and CTLA-4 in intratumoral Tregs and CD8^+^ T-cells, leading to immune cell exhaustion and a subsequent immunosuppressive microenvironment [73]. In addition, VEGF was also reported to inhibit the maturation of dendritic cells, which impeded T-cell priming and activation against tumors [72]. Additional evidence suggests that angiogenic factors such as VEGF, PGE2, and IL-10 promote the surface expression of Fas ligand (FasL) on the tumor endothelium, creating an immunosuppressive microenvironment by selective of CD8^+^ T-cells [74]. FasL is highly cytotoxic, and co-cultures of T-cells and FasL-expressing endothelial cells induced T-cell apoptosis in vitro and inhibited T-cell infiltration in mouse tumors, a phenotype reversed by the pharmacological inhibition of FasL [74]. In these studies, this immunosuppressive phenotype was also attenuated by treatment with anti-angiogenic therapies, which interferes with PD-1/PD-L1 and Fas ligand–receptor axes in the TME [72,73,74]. These results should inform the rationale for combining anti-angiogenic and immunotherapies to treat certain solid tumors.

Another study reported that co-cultures of liver cancer cells or patient xenografts organoids with endothelial cells upregulated CXCL16, IL-8, MCP-1, and TNF-α expression and secretion. This recapitulated the known pro-inflammatory environment characterized by infiltration of tumor-associated macrophages previously reported in liver cancer [75]. Endothelin-1, a potent vasoconstrictor secreted by endothelial cells, is a critical regulator of immune-cell trafficking. Co-expression of endothelin-1 and its receptors ETAR and ETBR correlated with microvessel density in ovarian and breast carcinoma [76,77]. The binding of endothelin-1 to ETBR attenuated the adhesion and infiltration of T-cells to the endothelium regardless of stimulation with the inflammatory cytokine TNF-α [76]. Similar findings were reported in glioblastoma, where ETBR-negative vessels had elevated infiltrating cytotoxic T-cells and low numbers of regulatory T-cells [78]. The pharmacological inhibition of ETBR promoted T-cell homing to tumors and augmented the effectiveness of cancer vaccines in mouse models [76]. The angiocrine factor angiopoietin-2 (ANGPT2) is secreted by endothelial cells following angiogenic or inflammatory stimuli and has been implicated in immunosuppression [79,80]. ANGPT2 inhibits the cytotoxic role of monocytes by blocking the secretion of TNF-α. In addition, ANGPT2 drives the upregulation of adhesion molecules, which increases the interactions between endothelial cells and leukocytes and promotes the recruitment of regulatory T-cells and myeloid-derived suppressor cells that induce immunosuppression [79]. In human and murine melanomas, ANGPT2 upregulation was associated with cytotoxic T-cell exclusion and immune evasion, a phenotype reversed upon pharmacological inhibition of this factor. Interestingly, the blockade of ANGPT2 signaling increased cytotoxic CD8^+^ T-cell infiltration into the tumor and improved the response to immunotherapy in melanoma mouse models [80]. Similar roles of ANGPT2 in tumor immunosuppression have been reported for non-small cell lung cancer and glioblastoma [81,82]. 

## 10. Conclusions and Future Perspectives

Evidence shows that crosstalk between endothelial, tumor, and stromal cells at primary and metastatic sites shapes tumor progression through different angiocrine factors. This increases the possibility of developing novel anti-angiocrine therapies for cancer treatment. The success of these new therapies will depend on a comprehensive understanding of the molecular dichotomy of endothelial cells in angiogenesis and angiocrine signaling. First, there is limited knowledge of how pre-treatment angiocrine signatures regulate the response to approved anti-angiogenic therapies. Second, we lack an understanding of the post-treatment angiocrine signature and its subsequent effect on the response to treatment and metastasis. Third, while single-cell studies on endothelial cell heterogeneity have elucidated some mechanisms that likely contribute to the resistance to anti-angiogenic therapies, there are limited data about how this heterogeneity facilitates tumor progression. A fundamental challenge of these bulk and single-cell transcriptomic studies is translating their results into diagnostic and therapeutic solutions, given the mixed outcomes of these findings. These limitations present new avenues for research to support developing anti-angiocrine therapies. 

Nevertheless, the multiple studies linking angiocrine factors to tumor growth, metastasis, therapy resistance, dormancy, the establishment of a pre-metastatic niche, and immune evasion point to an active endothelium that needs to be targeted. For example, the endothelium secretes TSP-1, keeping the extravasated metastatic breast cancer cells dormant [61,62]. In the later stages of dormancy, the sprouting endothelium secretes TFG-β and periostin, which awakens the dormant cancer cells, facilitating colonization of the secondary organ [61]. Moreover, angiocrine signaling is tumor-specific, as a single factor may regulate different functions. For example, ANGPT2 has been implicated in tumor metastasis, escape from dormancy, and immune evasion in the liver, breast, and melanoma [40,65,80]. While targeting angiocrine factors in combination with existing therapies has been effective, at least in pre-clinical cancer models, further in vivo studies are required to develop and understand drugs’ mechanisms of action against these factors. However, the spatiotemporal landscape of angiocrine signaling in solid tumors and the tumor-specificity of individual angiocrine factors present a complex barrier [2,15,26]. Additional in-depth investigations are required to unravel the complex angiocrine landscape during tumor progression before conducting clinical trials in patients. Despite these caveats, the continued discovery of the heterogeneity of endothelial cells, the identification of alternative factors that regulate angiogenesis, and the understanding of tumor-specific angiocrine signatures provide a solid foundation for developing unique therapies targeting the tumor vasculature. 

## Figures and Tables

**Figure 1 cells-12-02510-f001:**
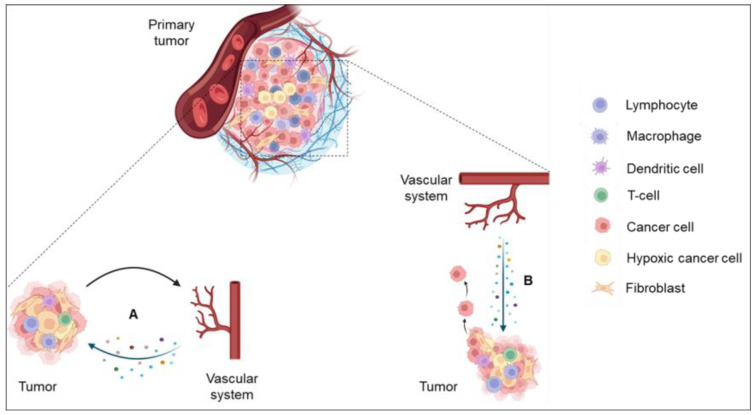
Types of angiocrine signaling in the tumor microenvironment. (**A**) The tumor or peripheral endothelium responds to micro-environmental cues from tumor, immune, and stromal cells by secreting angiocrine factors, which act in a paracrine manner to confer multiple growth advantages to the tumor. (**B**) Endothelium-initiated paracrine signaling, devoid of activation by micro-environmental signals, induces tumor invasion and survival.

**Figure 2 cells-12-02510-f002:**
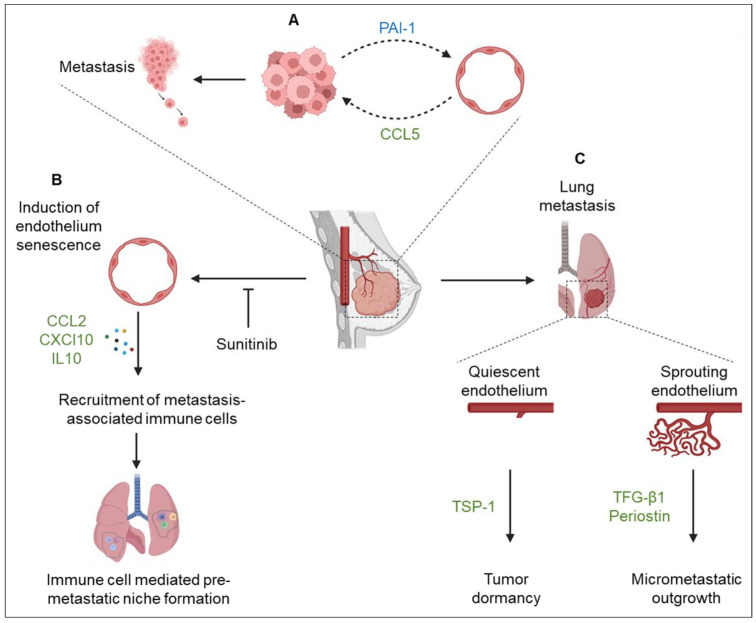
Angiocrine signaling in breast cancer. (**A**) Tumor inflammation activates the secretion of PAI-1 by tumor cells, stimulating the endothelium to secrete CCL5, which acts in a paracrine manner on tumor cells to facilitate metastasis. (**B**) Treatment of breast tumors with sunitinib, an anti-VEGFR inhibitor, induces endothelium senescence and subsequent secretion of angiocrine factors CCL2, CXCL10, and IL-10. These factors recruit metastasis-associated immune cells, promoting pre-metastatic lung niche formation. (**C**) The status of the endothelium determines the fate of metastatic breast cancer in the lungs. While quiescent endothelium secretes thrombospondin 1, facilitating tumor dormancy, sprouting endothelium secretes TGFB-1 and periostin, activating micrometastatic outgrowths.

**Figure 3 cells-12-02510-f003:**
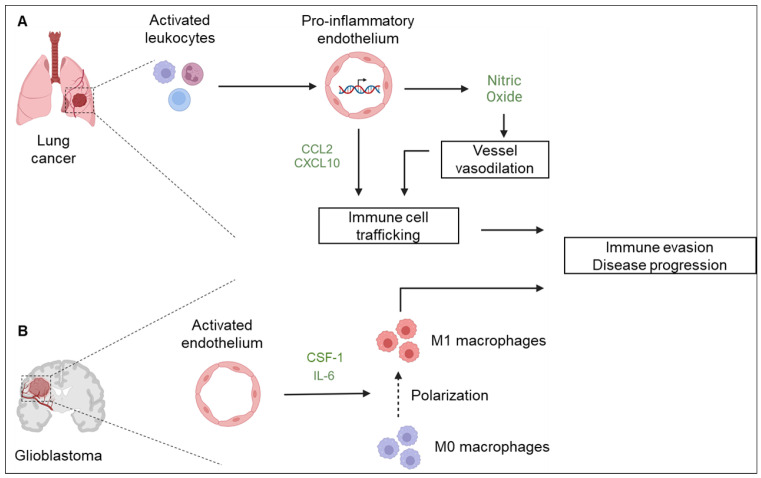
Angiocrine signaling promotes immune evasion. (**A**) In lung cancer, activated leukocytes induce a pro-inflammatory transcription program in the endothelium, leading to nitric oxide, CCL2, and CXCL10 secretion. Nitric oxide regulates vasodilation, while CCL2/CXCL10 collectively promotes immune-cell trafficking, leading to immune evasion and disease progression. (**B**) In glioblastoma, activated endothelial cells secrete IL-6 and CSF-1, which promotes macrophage polarization, driving immune evasion and disease progression.

**Table 1 cells-12-02510-t001:** List of FDA-approved anti-angiogenic therapies for cancer.

Drug	Molecular Targets	FDA Approved Indication
Lenvatinib	c-Kit, VEGFRs, PDGFRa, RET, FGFRs	Thyroid cancer, renal cell carcinoma
Cabozantinib	AXL, cMet, Tie2, VEGFRs	Medullary thyroid cancer, renal cell carcinoma
Axitinib	c-Kit, PDGFRs, VEGFRs	Renal cell carcinoma
Sorafenib	FLT3, KIT, RET, RAF, PDGFRs, VEGFRs	Renal cell carcinoma, thyroid cancer, hepatocellular carcinoma
Vandetanib	EGFR, RET, VEGFRs	Medullary thyroid cancer
Pazopanib	c-Kit, FGFR1, FGFR2, PDGFRs, VEGFRs	Soft tissue sarcoma, renal cell carcinoma
Aflibercept	VEGF-A, VEGF-B, PGF	Metastatic colorectal cancer
Sunitinib	CSF1R, FLT3, PDGFRs, RET, VEGFRs	Pancreatic neuroendocrine tumors, renal cell carcinoma
		Gastrointestinal stromal tumors
Ramucirumab	VEGFR2	Metastatic colorectal cancer, gastric adenocarcinoma
		Non-small cell lung cancer
Regorafenib	KIT, TIE2, RET, RAF, FGFRs, PDGFRs, VEGFRs	Gastrointestinal stromal tumors, hepatocellular carcinoma
		Metastatic colorectal cancer
Bevacizumab	VEGF-A	Metastatic colorectal cancer, lung cancer, renal cell carcinoma
		Ovarian cancer, cervical cancer

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
