# Peer review of "Tumor Angiocrine Signaling: Novel Targeting Opportunity in Cancer"

_cells, 2023, doi:10.3390/cells12202510_

Round 1

Reviewer 1 Report

This is a wonderful manuscript explaining the angiogenesis signaling and factor that regulate it in the TME.

Comments: 

1. Author also described the approved anti-angiogenesis therapies. I suggest summarize the available therapy in a table.

2. There are many single cell resolution studies done those found the heterogeneity  of endothelial cells in TME. Being its importance in therapy, author should also described this fact in manuscript.  

NA

Author Response

Response to Reviewer 1

Reviewer Comments: 

  1. Author also described the approved anti-angiogenesis therapies. I suggest summarize the available therapy in a table.

Response: We have added a table with a list of FDA-approved angiogenesis therapies (Page 18 of the revised manuscript).

  1. There are many single cell resolution studies done those found the heterogeneity of endothelial cells in TME. Given its importance in therapy, author should also described this fact in manuscript.  

Response: We have included a paragraph about the heterogeneity of endothelial cells based on single cell resolution studies (Page 4 of the revised manuscript)

Reviewer 2 Report

I am pleased to be involved in the peer review of this excellent review. This reviewer has the following concerns:

#1 It is necessary to provide a summary of angiogenesis inhibitors. The objective of this manuscript appears to be to deepen our understanding of endothelial signaling in order to develop effective therapeutic strategies. For this purpose, I recommend creating a new table that lists currently approved angiogenesis inhibitors.

#2 The description concerning the PD-1/PD-L1 axis is sparse. It has been reported that VEGF-A (vascular endothelial growth factor)-A, which is produced by both tumor cells and endothelial cells, plays a role in the formation of an immunosuppressive microenvironment by enhancing the production of inhibitory checkpoint molecules like PD1 (1).

The mention of PD-1 should likely be added within the main text of the paper.

Ref 1: Voron T, Colussi O, Marcheteau E, Pernot S, Nizard M, Pointet AL, Latreche S, Bergaya S, Benhamouda N, Tanchot C, et al. VEGF-A modulates expression of inhibitory checkpoints on CD8+ T cells in tumors. J Exp Med. 2015;212(2):139–148. doi: 10.1084/jem.20140559.

N/A

Author Response

Response to Reviewer 2

Reviewer Comments: 

  1. It is necessary to provide a summary of angiogenesis inhibitors. The objective of this manuscript appears to be to deepen our understanding of endothelial signaling in order to develop effective therapeutic strategies. For this purpose, I recommend creating a new table that lists currently approved angiogenesis inhibitors.

Response: We have added a table with a list of FDA-approved angiogenesis therapies (Page 18 of the revised manuscript).

  1. The description concerning the PD-1/PD-L1 axis is sparse. It has been reported that VEGF-A (vascular endothelial growth factor)-A, which is produced by both tumor cells and endothelial cells, plays a role in the formation of an immunosuppressive microenvironment by enhancing the production of inhibitory checkpoint molecules like PD-1. The mention of PD-1 should likely be added within the main text of the paper.

Response: We have included a paragraph about the immunomodulatory effect of VEGF in tumors and how its drives an immunosuppressive microenvironment (Page 11 of the revised manuscript)